# Influence of Heat Stress on Poultry Growth Performance, Intestinal Inflammation, and Immune Function and Potential Mitigation by Probiotics

**DOI:** 10.3390/ani12172297

**Published:** 2022-09-05

**Authors:** Rafiq Ahmad, Yu-Hsiang Yu, Felix Shih-Hsiang Hsiao, Chin-Hui Su, Hsiu-Chou Liu, Isabel Tobin, Guolong Zhang, Yeong-Hsiang Cheng

**Affiliations:** 1Department of Biotechnology and Animal Science, National Ilan University, Yilan 26047, Taiwan; 2Ilan Branch, Livestock Research Institute, Yilan 268020, Taiwan; 3Department of Animal and Food Sciences, Oklahoma State University, Stillwater, OK 74078, USA

**Keywords:** thermoregulation, reproduction, strategies, body temperature, poultry

## Abstract

**Simple Summary:**

The poultry industry sustains severe economic loss under heat stress conditions. Heat stress adversely affects the productivity, physiological status, and immunity of birds. To date, several mitigation measures have been adopted to minimize the negative effects of heat stress in poultry. Nutritional strategies have been explored as a promising approach to mitigate heat stress-associated deleterious impacts. Of these, probiotic feeding has a strong potential as a nutritional strategy, and this approach warrants further investigation to improve thermotolerance in poultry.

**Abstract:**

Heat stress has emerged as a serious threat to the global poultry industry due to climate change. Heat stress can negatively impact the growth, gut health, immune function, and production and reproductive performances of poultry. Different strategies have been explored to mitigate heat stress in poultry; however, only a few have shown potential. Probiotics are gaining the attention of poultry nutritionists, as they are capable of improving the physiology, gut health, and immune system of poultry under heat stress. Therefore, application of probiotics along with proper management are considered to potentially help negate some of the negative impacts of heat stress on poultry. This review presents scientific insight into the impact of heat stress on poultry health and growth performance as well as the application of probiotics as a promising approach to alleviate the negative effects of heat stress in poultry.

## 1. Introduction

The utmost important goal of the global poultry industry is to ensure a steady supply of eggs and meat to consumers. The poultry industry has produced superior-quality chickens (*Gallus gallus domesticus*) through intensive farming practices, modern equipment, extensive and balanced ration formulas, and other innovative technologies [1]. In the near future, broiler production is expected to grow sharply due to rapid and sustainable production in tropical and sub-tropical regions worldwide. Poultry meat is a rich and elementary source of nutritional proteins, vitamins, and minerals, and is low in saturated fatty acids [2,3,4]. Likewise, poultry eggs have a fundamental importance and are a cheap source of animal protein [5]. In addition to proteins, minerals, and vitamins, poultry eggs contain high amounts of zeaxanthin and lutein, which contribute to eye health [6].

According to a survey by the Food and Agriculture Organization in 2012, 103.5 million tons of chicken meat is produced around the world annually, contributing 34.3% to overall global meat production [7]. It is apparent that, within the livestock sector, the poultry industry has gained an active and leading role in most regions of the world [8]. In fact, due to robust growth and basic importance, poultry meat and egg consumption has doubled in the past decade [9]. As a result, the poultry industry has worked diligently to compensate for the increase in consumer demand. For example, at the start of the 19th century, egg production was documented at 150 eggs per hen per year, and presently hens are laying around 300 eggs annually [10]. This large increase in the production rate of broilers and layer hens has resulted in an increase in metabolic rate correlating with a rise in heat production causing these genetically superior birds to be highly susceptible to heat stress [11].

As the environmental temperature is steadily increasing worldwide, heat stress is considered a serious threat to the poultry industry in many countries. The Intergovernmental Panel on Climate Change reported a 1.53 °C increase in average environmental temperature between 2006–2015 compared to 1850–1900. It has nearly doubled in comparison to the average global temperatures since the pre-industrial period. This steady increase was anticipated to result in novel, hot climates, particularly in tropical regions. Climate change has an adverse impact on food security because of its detrimental consequences on agricultural crops and livestock [12]. Furthermore, the temperature is regarded as the most important environmental determinant among all bioclimatic variables known to cause stress. Nowadays, heat stress is one of the major climatic issues affecting the production, reproduction, and growth performance of various livestock species and poultry [13]. Poultry under high ambient temperatures develop physiological, behavioral, and immunological responses, which directly or indirectly adversely affect their performance [3].

According to a survey in 2003, the poultry industry faces $128 million in economic losses per year [14], and due to the continuous rise in global temperature, this number is suspected to increase in the near future. Poultry are most vulnerable to heat stress because their body is covered by feathers and they lack sweat glands, leaving them unable to release body heat [15]. Heat stress can be defined as a state where poultry cannot maintain a balance between body heat production and dissipation. There are certain factors that contribute to heat stress such as environmental temperature, moisture level in air, radiation, and air moment. In particular, high environmental temperatures and high air moisture levels have the largest impact on heat stress in poultry [16]. Certain factors play critical roles in the ability to bear high ambient temperatures above the comfort zone such as stage of life, sex, breed attributes, general body weight, growth and performance rate, physiological status, feeding intervals, and relative humidity [17]. On average, the chicken body temperature ranges from 41–42 °C. In order to achieve a maximal growth rate, the bird must remain in the thermoneutral zone, which is between 18–21 °C [18].

Heat stress can be classified as acute or chronic. Acute heat stress can be defined as a sudden rise in temperature and humidity for short intervals of time [19]. A prolonged and continued period of high environmental temperature along with elevated humidity results in chronic heat stress [20]. Typically, in birds, heat stress can be categorized as follows: acute heat stress ranges from 27–38 °C for 1–24 h, moderate heat stress ranges from 27–38 °C for a duration of 7 days, chronic heat stress is high ambient temperatures of 38–50 °C for a prolonged period of 7 days or more [21]. Heat stress detrimentally affects the growth and productive efficiency, economic traits, reproductive capability, and well-being of poultry. Commonly, in broilers, a body core temperature beyond the comfort zone adversely affects and agitates morphological homeostasis and represses the activity of intestinal immunity and digestion, consequently leading to gastrointestinal tract inflammation [22]. As a result, the health status of the bird is gradually decreased, and the mortality rate is increased. The overall health of the intestinal tract plays a fundamental role in broilers and layers in terms of meat and egg production, respectively, as the intestinal tract has a direct influence on general poultry welfare and production rate [22].

Therefore, a better understanding of the impacts of heat stress on poultry welfare, performance, and production is required to develop probiotic feed supplements to mitigate heat stress to promote avian welfare, sustainable performance, and production. This paper reviews the current knowledge and research progress on the consequences of heat stress on poultry performance and poultry welfare, and also identifies some of the potential probiotic nutritional interventions to alleviate heat stress consequences to improve poultry welfare.

## 2. Thermoregulation of Birds in Response to Heat Stress

Poultry manage and maintain their body temperature by balancing metabolic heat production and dissipation during changes in environmental conditions. Several factors such as extreme heat, climate changes, temperature fluctuations, and increased moisture levels lead to heat stress in poultry [20]. Poultry develop and adopt certain morphological, physiological and behavioral traits under heat stress conditions to maintain their normal body temperature [17].

Poultry are said to be under heat stress conditions during an imbalance between body thermogenesis and heat dissipation [23]. Thermally stressed animals reduce heat production by lowering feed intake, which has a detrimental effect on performance productivity and well-being. Excessive increases in environmental temperature have detrimental consequences on the morphological, behavioral, growth performance and productive features of poultry [24,25,26]. All breeds and ages of poultry generate heat. Poultry are comfortable and perform basic life processes from 23.9 °C to 26.7 °C [27].

The key sign of heat stress development in poultry flock is the rise in conservation of energy cost. As the environmental temperature increases above the comfort zone, the body will adjust to lose heat through radiation via different parts of body such as feet, comb, and wattles. Birds will also modify their behavior and may perform wing spreading and panting. Since most of the body of poultry is covered with feathers, heat dissipation through the wattle, head, comb, and feet is minimal. However, panting is the key procedure acquired by poultry birds to undergo heat loss during heat stress conditions [28]. Air sacs play a vital role in the transfer of body heat to the ambient environment using a respiratory evaporative mechanism to decrease and maintain the normal body temperature [29]. In the panting mechanism, the air sacs become more essential as they promote body surface air dissemination to minimize body heat dissipation by an evaporative mechanism. However, extreme, and uncontrolled panting under heat stress conditions in poultry contributes to a decrease in the availability of calcium and carbon dioxide pressure. As a consequence, blood pH levels increase, leading to respiratory alkalosis to cause bone deterioration and lameness24.

In addition, poultry under heat stress conditions have reduced feed intake, water consumption, body moments, and consequently become depressed, dull and lethargic [30]. From a practicable perspective, it is crucial to consistently and properly monitor water utilization, feed intake, sleep interval, movement, and other behavioral activities when conditions are favorable for the development of heat stress in commercial poultry production. The prompt decrease in water intake, reduced feed consumption, behavioral changes, and physiological appearance of the body are the best and key indicators of birds experiencing heat stress [31,32]. The effects of heat stress on growth, behavioral, and physiological traits are shown in Figure 1.

## 3. Influence of Heat Stress on Growth Performance

The hypothalamus is interconnected with the regulation and maintenance of vital activity responsible for normal bodily functions such as metabolic heat production, water consumption, general motion, and appetite. When chickens are exposed to heat stress, a reduction in feed consumption is normally observed [33]. Remarkable progress has been achieved in understanding the complex interaction of the neuroendocrine signals, specifically the leptin neuropeptide Y (NPY) axis in appetite control. In animals, the most effective stimulant of feed consumption is leptin [34]. Leptin is a 16-kDa protein expressed in a variety of organs containing adipose tissue. Adipocytes are responsible for generating and providing leptin in sufficient quantities, which is primarily associated with obesity and secondarily associated with the general body weight of animals [34]. Leptin and adiponectin produced by adipocytes have a vital role in the supervision and control of feed intake [34]. In chickens raised in thermos-neutral conditions, leptin suppresses *HSP70* gene expression [35]. Leptin and adiponectin signaling are enhanced by heat-stressed conditions in mice (*Mus musculus*) [36].

The effect of leptin and adiponectin in poultry flocks under heat stress conditions is yet to be determined and requires more investigation. Several distinct hormones have fundamental activity in the neuro-hypothalamic axis in sustaining feed intake and energy homeostasis. The most documented and well understood hormones are ghrelin and cholecystokinin (CCK). Poultry subjected to heat stress conditions at 32 °C for 14 days had increased gene expression of ghrelin and CCK [37]. These findings suggest that an increase in ghrelin and CCK is linked to a reduction in feed consumption and may be an accountable cause of decreased appetite in heat-stressed chicks. 

## 4. Influence of Heat Stress on Reproductive Performance

### 4.1. Female

Heat stress affects the ovulation rate in laying and breeder chickens resulting in lower reproductive performance [38,39]. It may be due to decreased secretion of gonadotropin-releasing hormone, luteinizing hormone, and follicle stimulating hormone [40]. Additionally, an excessive rise in temperature alters hypothalamic reproductive activity regulation in laying chickens and decreases luteinizing hormone concentrations in the bloodstream. This phenomenon is caused by hypothalamic dysfunction [41,42]. Heat stress induces oxidative stress and alters the physiological performance of the small yellow follicles, ovaries, and oviducts in laying quails (*Coturnix coturnix*), hens (*Gallus gallus domesticus*), and ducks (*Anatidae*) resulting in a considerable reduction in the relative weight of the ovaries, oviducts, and the total count of large follicles. As a result, egg production is reduced, and in extreme conditions, sterility may develop [43,44]. In a study of 96 laying hens (60 weeks of age), under heat-stressed conditions at 34 °C and 65% relative humidity during a 12-day experimental period, reduced feed intake and decreased egg production was recorded [45]. Decreased feed consumption subsequently leads to impaired egg production and hyperthermia. However, total egg weight remained unaffected in laying hens subjected to heat stress environmental conditions [26,45,46].

Birds feel comfortable at 24 °C and can function normally up to 26 °C. However, having broilers raised in environmental temperatures above 27–29 °C causes decreased feed intake and increased water consumption, and decreases in feed conversion ratio and weight gain, while in layer and breeder flocks, egg production decreases at these temperatures [27]. At 30–35 °C, a crucial decrease in egg production and eggshell quality has been documented. A high ambient temperature detrimentally alters feed intake and the feed conversion ratio in poultry [47]. Egg production, egg size, eggshell quality, hatchability, and fertility are decreased in layers and breeder flocks when exposed to heat stress conditions [48,49,50]. Increases in serum cholesterol, serum glucose concentration and decreases in feed intake egg production, egg weight, eggshell thickness, eggshell weight, Haugh unit, and eggshell strength have been documented under excessive ambient temperature conditions (37 ± 2 °C) compared to thermoneutral conditions (27 ± 2 °C) [51]. In Japanese quails (*Coturnix japonica*), the overall fertility percentage in thermoneutral conditions (23.8 ± 0.7 °C) was 84.8%, while under heat stress conditions (35.8 ± 0.6 °C), fertility was reduced to 78.9%. Similarly, the hatchability percentage in the thermoneutral group (23.8 ± 0.7 °C) was 80.2%, whereas in heat stress conditions, hatchability dropped to 74.1% [45].

### 4.2. Male (Heat Stress on Spermatogenesis)

Heat stress has comprehensive and diverse consequences on testes and testicular function in different animals. Different cell types have varying responses to heat stress. The Leydig cells found in the testes are not directly affected by heat or are minimally affected. However, the Sertoli cells may be the primary target of heat stress [52]. These cells may be impacted due to their position in the seminiferous epithelium. The Sertoli cells deliver all nutrients to the germ cells and their growth is regulated by the environmental conditions. The spermatogonia and preleptotene spermatocytes are influenced by the Sertoli cell [53].

Fertility is one of the most important characteristics in breeder flocks that needs to be measured, especially under heat stress. Thermally stressed chickens, ducks, and quails have poor physiological well-being statuses. There is a complex relationship between the environment and reproductive performance since heat stress can alter gonad sensitivity to metabolic hormones through the alteration of receptor signaling 14.

Male breeders are more affected by heat stress than female breeders in terms of infertility [54]. An excessive rise in ambient temperature above the thermoneutral zone prompts an increase in lipid peroxidation due to reactive oxygen species generation, which damages testes and eventually has deleterious repercussions on the seminal parameters. The basic seminal parameters are semen volume, testes weight, sperm concentration and motility, sperm viability, spermatids, spermatocytes, and spermatogonia [55]. These are influenced by several factors such as temperature, pH, and ion concentration. Therefore, suboptimal conditions can ultimately result in poor-quality spermatozoa and infertility [55,56]. In the initial phases, a temperature rise promotes testicular growth, semen quantity, and concentration; however, excessive increases in temperature suppresses the reproductive capabilities of broilers [57]. Five Japanese male quails (*Coturnix japonica*) out of 45 experiencing heat stress conditions had significantly lower reproductive efficiency [58]. As the temperature exceeded the thermal comfort zone, consequently semen quality and quantity were reduced.

Furthermore, hens inseminated with sperm taken from heat-stressed roosters produce less fertile eggs due to the lower egg production. During summer season, semen collected from roosters was found to have certain spermatozoa deformities including deformed heads, a split mid-piece, and cytoplasmic droplets [57]. In a 28-day experimental phase, reductions in the abundance of all spermatocyte cells, including spermatogonia, spermatocytes, spermatids, and spermatozoa, were documented in quails in heat stress conditions (34 °C, 8 h/day, 60–65% RH) [59]. The semen collected from heat-exposed roosters had a shorter lifespan in the uterovaginal junctions of the hen. According to several studies, artificial insemination success rates are significantly higher for the semen specially collected at the morning intervals of the day [40,57]. The effect of heat stress on poultry performance are shown in Table 1.

## 5. Influence of Heat Stress on Intestinal Health

In the digestive tract, the intestine plays a very important role as most of the nutrient absorption occurs here. The poultry intestine harbors a diverse community of microorganisms, which helps in the breakdown of complex nutrients to simple forms that can be easily digested and absorbed [66,67,68,69]. The arrangement and adherence of epithelial cells of the gut play a very important role in absorption of nutrients and protection of the body from the invading, harmful microorganisms into the blood stream [70]. Environmental factors, especially heat stress, have a very important effect on the microbial population of the poultry gut. Heat stress alters the microbial population and the activities performed by these microbiota, but the exact mechanism of its alteration is still unknown [71,72]. Heat stress adversely affects the morphological structure of the duodenum, jejunum, and ileum of the small intestine. During heat stress, the intestinal morphology is adversely affected, including as changes in relative weight, villi height, villi surface area, crypt depth, and surface area of the epithelial and immunoglobulin A-secreting cells [56,73,74].

During heat stress, feed intake is reduced and water intake is enhanced, which has a negative impact on the absorption of nutrients being produced by the microbiota in the intestinal lumen. Additionally, the secretion and motility of the intestine increases and nutrient absorption decreases [75,76]. Production of digestive enzymes in the intestinal lumen is decreased by the heat stress, which adversely affects the intestinal mucosa leading to oxidative stress and inflammation. Heat stress decreased populations of beneficial bacteria (*Lactobacillus* and *Bifidobacterium*) in the intestine which was replaced by harmful bacteria like *Coliforms* and *Clostridium*. These changes lead to a decrease in digestibility of the feed and low availability of nutrients for absorption from the intestinal mucosa to the blood [73,77,78].

As the number of harmful microbes in the intestine increases it can lead to leaky gut syndrome [79]. Physiological changes resulting from heat stress allow the pathogenic bacteria to enter the bloodstream by crossing the intestinal lumen to cause septicemia. In this manner, heat stress leads to lower performance, less egg production, low meat yield, retarded immunity status, and low reproductive performance [79]. The impact of heat stress on the intestinal microbial population is of interest in recent studies, and there is a need to study the impact of heat stress on the microbial population in more depth. More advanced methods and technologies are needed to study the impact of heat stress on gut microbes and the response of the body to heat stress, especially in poultry. The effect of heat stress on intestinal health and immune system are shown in Figure 2.

## 6. Influence of Heat Stress on Immunity Function

For optimal performance, the chicken requires a vigorous immune system. The immune system has different lymphoid organs around the body. There are many associated with the intestinal lumen, including different lymphoid cells present in the epithelial mucosa (intraepithelial lymphocytes) and the lamina propria, as well as specialized lymphoid structures, such as Meckel’s diverticulum, Peyer’s patches, bursa of Fabricius and cecal tonsils [32,80]. The intestinal immune cells and gut neurons share information with each other to tackle any stressor through combined actions [81]. Different immune cells such as lymphocytes, monocytes or macrophages, and granulocytes have receptors for many neuroendocrine products of the HPA and sympathetic-adrenal-medullary axes, such as corticosterone and catecholamines, which can affect cellular trafficking, proliferation, cytokine secretion, antibody production, and cytolytic activity [82]. Different studies have shown the effects of heat stress on the immune system of poultry [33,45]. All studies performed on broilers and laying hens have shown immunosuppressing effects when exposed to heat stress.

During heat stress different cells, like intraepithelial lymphocytes and immunoglobulin A-secreting cells, are adversely affected. Heat stress also reduced the antibiotic response and phagocytic activity of macrophages 33. Heat stress deteriorated the cell membrane by oxidative changes, caused atrophy of lymphoid organs like the thymus, bursa, and spleen, and decreased the volume of lobules of the thymus, lymphoid follicles, and medulla/cortex ratio [83,84,85]. As the immune system is under pressure due to heat stress, the harmful bacteria in the crop and intestine start to grow and colonize causing morphological changes in the intestinal lymphoid cells. These pathogens are presented to the immune cells by the antigen-presenting system and activate the expression of pro-inflammatory cytokines to tackle the invading pathogens [31,86]. In conclusion, heat stress alters the immune response of the body to invading pathogens by changing the Toll-like receptors and pro-inflammatory cytokines in poultry.

## 7. Influence of Heat Stress on Triiodothyronine and Thyroxine

The hypothalamic–pituitary–adrenal axis and the hypothalamic–pituitary–thyroid axis are two main thermoregulatory mechanisms of the hypothalamus in homeothermic animals including poultry [87,88]. The thyroid gland secretes ample amounts of T4 hormone (thyroxine). For the appropriate activity of the thyroid, T4 should be transmuted to T3 (3,5,3′-triiodothyronine). Thyroid hormones (T3 and T4) are involved in regulating homeostasis and nutritional digestion, which are affected by internal and external stimuli [89]. Synchronization of thyroid gland activity, such as T3 and T4 concentrations, is important for the regulation of body temperature and homeostasis through energy metabolism in poultry [90]. The thyroid gland has a central role in adjusting to high ambient temperatures because thyroid hormones (T3 and T4) have a vital role in controlling body metabolic rates of birds throughout growth and egg production phases [91].

Increases and decreases in plasma corticosterone and circulating thyroid hormones such as T3 and T4, respectively, develop due to excessive rises in ambient temperature. This change detrimentally affects the normal action of the neuroendocrine system, and as a result, the hypothalamic–pituitary–adrenal axes (HPA) become activated [83,92]. These different deleterious effects lead to debilitated metabolic function [18] and, conclusively, decreased weight gain. In broilers, decreased T3 blood concentrations have been documented for birds experiencing excessive heat stress at different conditions such as 36 °C for 4 to 6 h daily at age 22 to 42 days [93], 38 °C for 3 h daily at age 35 to 40 days [94]; and 38 °C for 1 h at day 40 [95]. A decreased blood T3 concentration avoids deleterious effects of catabolism in broilers in heat stress conditions.

Alternatively, [96] mention that broilers under acute heat stress at a temperature of 50 °C for 1 h at five days of age have decreased blood serum T3 and T4 concentrations. On the other hand, [97] expressed that serum T3 and T4 concentrations in broilers experiencing thermal stress at 41 °C for 4 to 6 h on day 4 remain unaffected. An increase in T3 and a continuous decrease in T4 plasma concentrations have been found in Japanese quails under extreme environmental conditions (34–35 °C) during the first six hours of heat exposure [98]. In quails under acute heat stress (38 °C for 24 h), blood serum T3 concentrations were reduced in comparison to birds raised in thermoneutral conditions, 25 °C [99]. Reduced protein production is the consequence of the decrease in thyroxine concentrations, which leads to reduced body weight and daily gain under heat stress conditions. From the nutritional point of view, loss of appetite and decrease in feed consumption is accredited for the reduced growth production in birds under high ambient temperature [100]. These reports and studies reveal that thyroid hormones of broiler birds are detrimentally affected due to heat stress, which may vary by several factors such as intensity of stress, level of duration, status, and age of the bird.

Corticosterone, the primary glucocorticoid hormone in poultry, has central importance in the metabolism of adipose tissue, governing appetite, and body thermogenesis [101]. Excessive rises in glucocorticoid concentrations cause agglomeration of adipose brown tissue and high lipogenesis of white adipose tissue leading to impaired metabolism [100]. Furthermore, glucocorticoids maintain body temperature by hampering the conversion of T4 to T3 [102]. An increase in blood serum glucocorticoid concentrations under heat stress activates the gluconeogenesis pathway, which results in an elevation in blood glucose concentration. In addition, the synthesis of vasopressin and adrenocorticotrophic hormone was provoked in birds in heat stress conditions. Adrenocorticotropic hormone was produced in increased amounts under increasing stress challenges, which consequently led to aldosterone production [103].

## 8. Influence of Heat Stress on Heat Shock Protein

Heat stress detrimentally affects the immune system [104] by provoking the activation of heat shock proteins (HSPs), which results in poor energy and cellular respiration [105], causing suppression of broiler growth performance. Heat stress is the prime cause of high activation of heat shock factors and heat shock proteins in bird tissues. HSPs are intracellularly incorporated, excessively conserved and pervasive proteins that are activated by numerous stress factors [106]. Researchers have derived and documented that HSP activation is influenced by environmental stresses (such as heat or cold stress, oxidative stress, UV radiation, and heavy metals), morphological stresses (such as cell differentiation, caloric restriction, growth factors, hormonal stimulation, and tissue development), and immunological stress (such as parasitic or bacterial infection, inflammation, fever).

Elevated ambient temperature alters cellular activity, and consequently, multiple irregular cell biological events take place, which regulate cell metabolism, cell membrane function, and promote cell destruction. Furthermore, apoptosis and necrosis pathways lead to cellular death [107]. Based on cellular level action, HSPs have a vital and defensive role against stress factors and are highly accredited by their chaperone function. Chaperones have the potential to bind and avoid the irreversible aggregation of denatured proteins [108].

HSPs are extensively well known for their association with all living organisms under heat stress conditions. The expression of HSP mRNA and the concentration of HSPs are induced by heat stress. These HSPs recognize unfolded or misfolded proteins and either rectify or inhibit inappropriate protein folding [109]. On the basis of their molecular weights, HSPs are classified into HSP100s, HSP90s, HSP70s, HSP60s, HSP40, and small HSP families [110,111].

In general, HSP70s expression is regarded as a reliable biomarker for changes in the body temperature (greater than 38.6 °C) [112]. Hence, increased HSP70 concentration may coincide with improved heat stress tolerance because of enhanced cytoprotective effects and efficient protein folding under a heat stress environment [113,114]. A recent study has documented that HSP70 has a substantial cytoprotective role in the intestinal epithelium, along with a role in the establishment of the intestinal barrier [115]. Additionally, HSP70 strengthens and regulates normal intestinal tight junctions [116] and creates an effective intestinal barrier in the ileum of poultry under a thermal ambient environment [115].

Furthermore, HSP60s are manifested in the epithelium of the villus and crypt cells of the small intestine. The prime and fundamental role of HSP60s is to safeguard cells by regulating the mitochondrial protein assembling process under multiple stress factors. [117]. Upregulation of HSP60s has been attributed to intestinal mucosal structure inflammation [118] and has been involved in enterocyte restoration [110]. HSP60 is a well-known chaperone in cell mitochondria and regulates the mitochondrial bilayer membrane to counter cellular damage [119]. Thus, HSP60 plays a critical role to safeguard chicken (*Gallus gallus domesticus*) renal cells under heat stress conditions. It was reported in [120] that HSP70s mRNA expression was significantly upregulated for Muscovy and Pekin ducks in pituitary, adrenal, liver, and renal tissues and brain, pituitary, adrenal, heart, liver, spleen, and renal tissues under a heat stress environment (39 °C for 1 h). In broiler chickens, HSP70 concentrations and HSP70 mRNA expression were significantly higher in liver and hypothalamus under heat stress conditions at 32 °C daily for 9 h [90].

## 9. Mitigation of Heat Stress by Probiotics

Nutritional approaches can balance environmental distress during persistent heat stress. Feed additives have been supplemented to diets as part of a nutritional approach to mitigate the negative effects of heat stress in poultry [16]. Amongst feed additives, probiotics are of increasing interest to poultry nutritionists, as such additives have been reported to improve physiology, gut morphology and structure, and immune function, thereby enhancing performance and health of heat-stressed poultry [121].

Probiotics are defined as feed supplemented with live beneficial micro-organisms (including: *Bifidobacterium*, *Lactobacillus*, and *Streptococci*), yeast cultures (with *Candida* and *Saccharomyces* strains), and fungi (*Aspergillus awamori*, *A. niger*, and *A. oryza*), which may improve poultry performance, gut health, gut microbial balance, and immune response [122,123,124]. More recently, probiotics have attracted considerable attention in subsiding oxidative damage caused by heat stress in the poultry. Furthermore, the supplementation of probiotics improved growth performance, feed conversion ratio and immune response in broilers [60]. The addition of probiotics to the broiler diet markedly enhanced feed intake, daily gain, total body weight, intestinal absorptivity, and immunity [45]. Numerous researchers have reported the potential benefits of probiotics in improving gut morphology and integrity in heat-stressed birds [125]. A study showed that probiotics can reverse damaged villus-crypt structures in birds subjected to heat-stress as a means of controlling corticosterone concentrations [47] and excessive release of pro-inflammatory agents leading to intestinal tissue damage and increased tissue permeability [45]. The existence of the mucus layer is crucial to maintaining the bird’s intestinal integrity. According to a report, heat stress in quails reduced the number of mucus-producing goblet cells located in the villi of the ileum [125].

Probiotic supplementation (mannan-oligosaccharide and *Lactobacillus*) has been shown to increase the serum T3 and T4 concentrations in heat-stressed broilers [92,126]. Considering that the thyroid hormone plays a vital role in the synthesis and stimulation of various structural proteins, enzymes, and hormones, it is reasonable to expect that increased thyroid hormone concentrations following probiotic supplementation would improve digestion and metabolism in heat-stressed chickens [127]. A possible factor for increasing T3 and T4 concentrations in heat-stressed birds with probiotic supplementation is the reduced circulating concentrations of corticosterone, as elevated corticosterone concentrations may contribute to hypothyroidism [47]. Therefore, due to the high sensitivity of probiotics to environmental changes, it is very useful to provide probiotics in poultry diets which reduces the ability of the probiotic to reach the bird’s digestive tract and thus enhances their beneficial roles [123]. Furthermore, acute heat stress conditions can severely affect gastrointestinal tract structure, villus morphology, and intestinal epithelial integrity in laying hens [45,51,73], and damage gut integrity and cause immunosuppression in broilers [60,61]. *Bacillus* is of industrial importance for a number of reasons, including its excellent safety record, rapid growth rate resulting in short fermentation cycles, and ability to secrete proteins into the extracellular medium [128].

*Bacillus*-derived peptides have been shown to have antifungal, antibacterial, antitumor, antiviral, anti-amoeba, and anti-mycoplasmic activities [129]. Several species of *Bacillus*, such as *Bacillus licheniformis*, *Bacillus subtilis*, *Bacillus cereus* and *Bacillus clausii*, have been used as probiotic supplements in animal diets [130]. Ref. [131] reported that supplementation of dietary probiotics reduced the negative effects of oxidative stress on semen quality and increased breast meat weight in chronic heat-stressed broilers. Among aerobic bacterium the *Bacillus subtilis* has been found to be one of the most crucial with beneficial effects on poultry diets by inhibiting the growth of aerobic pathogens, thereby increasing efficiency of dietary protein in poultry. *Bacillus subtilis* has an enhanced potential to produce exogenous digestive enzymes, enhancing intestinal development, improving immune responses and function [123], improving egg internal quality, and reducing egg yolk cholesterol concentrations [132,133].

The facultative anaerobic bacterium, *Bacillus licheniformis*, has a beneficial effect on poultry diets by improving the intestinal absorption surface area and promoting the growth and proliferation of probiotics such as *Lactobacillus*, *Bifidobacterium*, and *Aspergillus awamori* [45,61]. In studies, *Bacillus licheniformis* was reported to maintain the goblet cell population in both the ileum and caecum of heat-stressed hens [45]. Similarly, *Lactobacillus*-based probiotics increased the number of goblet cells in the duodenum and jejunum of broilers subjected to heat stress conditions [134]. The possible mechanism reported so far by which probiotics increase the number of goblet cells is to modulate mucin mRNA expression and accelerate goblet cell differentiation [135]. In addition, egg weight, quantity and quality were significantly increased in laying quails fed a diet supplemented with probiotics [136].

It is therefore very important to state that supplementation of dietary probiotics may be involved in preventing pathogens [137], improving nutrient absorption and enhancing immunity, which ultimately translates into improved physical performance and disease resistance in heat-stressed poultry.

Nutritional approaches can balance environmental distress during persistent heat stress. Feed additives have been supplemented to diets as part of a nutritional approach to mitigate the negative effects of heat stress in poultry [16]. Probiotics increase the nutritive value of feed by converting less degradable compounds into more digestible forms. As a result, the gut microbial population shifts, the composition of the gut microbiota changes, and the immune system is modulated. Feed digestion is improved by producing hydrolytic enzymes (phytases, lipases, amylases, or proteases) that enhance nutrient intake or by stimulating the host to generate more digestive enzymes [138]. Probiotics may enhance the production of vitamins, exopolysaccharides, and antioxidants, thereby boosting the nutritional value of feed. Some probiotics may also regulate cholesterol metabolism [139]. The digestion and absorption of nutrients take place in the proventriculus and small intestines. Ileum and colon boost the fermentation of undigested proteins and carbohydrates. This is a critical step in digestion because undigested nutrients are a risk factor for dysbiosis [140]. The example of different types of probiotics and their application in poultry under heat-stressed conditions are shown in Table 2.

## 10. Conclusions and Future Perspectives

Heat stress has emerged as a serious threat to the poultry industry in many poultry-producing countries due to excessive increases in global temperature. Heat stress can affect poultry growth, gastrointestinal health, immune function, production status, and reproductive activity, thereby affecting poultry performance. Probiotics appear promising to mitigate detrimental effects in poultry raised under heat stress conditions, as probiotics can promote gut morphological activities, microbial ecology, physiological function, immune responses, and productivity in birds raised under heat stress conditions. However, further research is needed to investigate the molecular changes induced by probiotics as well as the interactions between probiotics, pathogens, and epithelial cells. This will necessarily involve metagenomic, proteomic, and metabolomic exploration. Elaborating these unknowns will give researchers a stronger insight into the role probiotics play in improving poultry health and growth.

## Figures and Tables

**Figure 1 animals-12-02297-f001:**
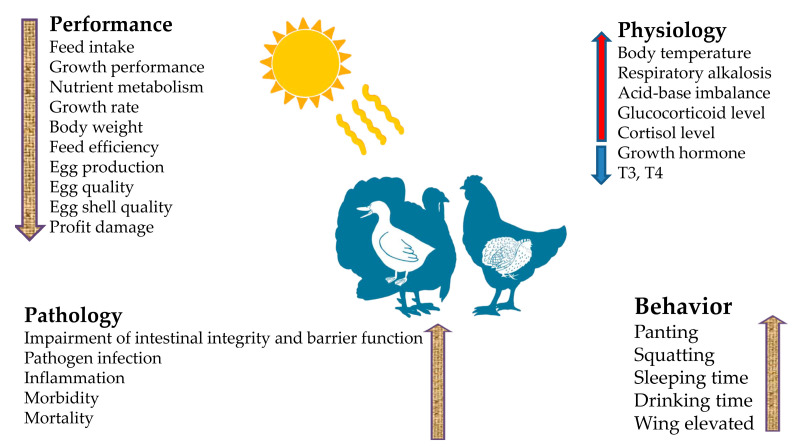
Effects of heat stress on growth, behavioral, and physiological traits.

**Figure 2 animals-12-02297-f002:**
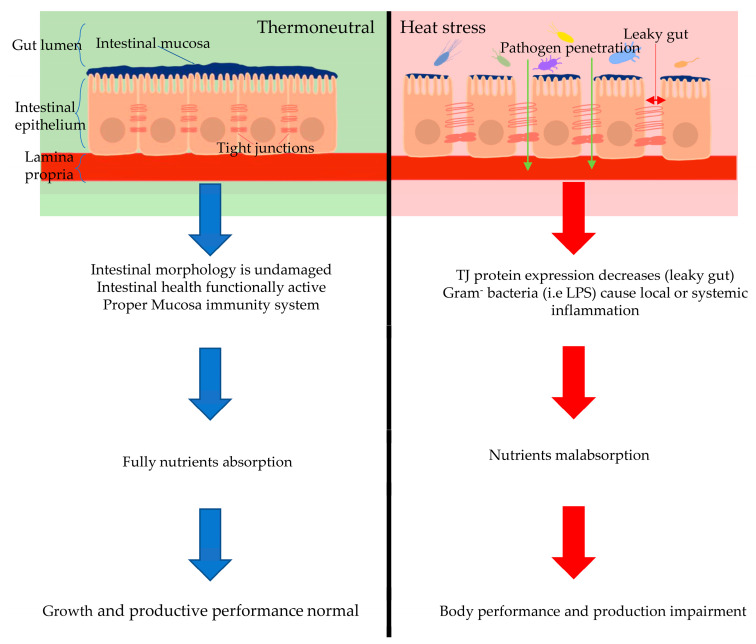
Effect of heat stress on intestinal health and immune system.

**Table 1 animals-12-02297-t001:** Effect of heat stress on poultry performance.

Species	Conditions	Outcome	Reference
Broiler chicken	32 °C	Excessive panting, elevated wings, ground squatting, standing, sleeping, sitting, and drinking, reduced feeding, body weight, AFI, and high FCR	[60]
Broiler chicken	30 °C	Decreased body weight, body gain weight, reduced growth hormone level, Insulin-like growth factor-1, increased cholesterol and glucose level, decreased villus height, crypt depth and villus surface area, and high FCR	[61]
Layers (Hen)	34 °C, RH 65%	Decreased feed intake, egg production, decreased villus height, crypt depth, and villus surface area High TNF-α, IL-1, IL-6, corticosterone	[45]
Layer	34 ± 2 °C	Decreased feed intake, low egg production, eggshell thickness, eggshell strength, and less egg weight	[51]
Layer (Hen)	35 °C, RH 65%	Decreased feed intake, weight gain, egg weight, and eggshell thickness	[26]
Japanese quails (12 weeks old)	35.8 ± 0.6 °C	Fertility 78.9% Hatchability 74.1% Poor eggshell quality, reduced eggshell thickness, eggshell weight, eggshell percentage	[43]
Japanese laying quail	34 °C	Feed intake decreased by 7.1% Bodyweight decreased by 7.7% Body weight gain decreased by 14.5% Egg production decreased by 23.3% Egg weight decreased by 14.3%	[62]
Egg-laying sheld duck (*Tadorna ferruginea*)	34 °C, RH 65%	Feed intake decreased by 11.9% Daily egg production decreased by 7.3% Egg weight decreased by 2.9% Decreased eggshell strength, eggshell thickness	[39,63]
Shanma duck Laying duck	34 ± 1 °C	Decreased feed intake, reduced egg weight, lower egg albumen height, and Haugh unit	[64]
Turkey	35 °C 35 °C	Decrease T3 (37.5%) Increase T4 (30%)	[65]

**Table 2 animals-12-02297-t002:** Example of different types of probiotics and their application in poultry to ameliorate intestinal morphology under heat-stressed conditions.

Probiotic Strains	Biological Performance under Heat Stress Conditions	Reference
*Bacillus subtilis* PB6	Increased the population of *Bifidobacterium* and *Lactobacilli* in the gut 3 × 10^7^ CFU/kg (broiler chicken) Increased villus height, crypt depth, and promoted surface area for absorption in duodenum and ileum (broiler chicken)	[64]
*Bacillus licheniformis*	Restored the impaired villus-crypt structure and maintained normal surface area in the small intestine (laying hens) Avoided the impairment and regulated the stability of the epithelial intestine (layer chickens)	[45]
Probiotic mixture (*B. licheniformis*, *B. subtilis*, *L. plantarum*	Reversed the reduced villus height, crypt depth, and surface area structure (broiler chicken) Increased beneficial bacterial population *Lactobacillus* and *Bifidobacterium* and decreased injurious bacterial population *Coliforms* in small intestine (broiler chicken)	[74]
Probiotic mixture (*L. pentosus ITA23* and *L. acidophilus ITA44)*	Increased the population of beneficial bacteria *Lactobacillus*, *Bifidobacterium*, *Enterococcus* (broiler chicken) Decreased the population of *E. coli*, respectively, in the gut (broiler chicken)	[141]

## Data Availability

Not applicable.

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
