# Peer review of "Influence of Heat Stress on Poultry Growth Performance, Intestinal Inflammation, and Immune Function and Potential Mitigation by Probiotics"

_animals, 2022, doi:10.3390/ani12172297_

Round 1
Reviewer 1 Report
In the current study, the authors have rviewed "Influence of heat stress on poultry growth performance, intestinal inflammation, and immune function and potential mitigation by probiotics". The authors have
Already there are a lot of review on the subject. The current study does not add anything to the existing knowledge of heat stress on poultry. Many important rerferences have been ignored. For me it is not understandable why a few parameters have been targeted. In the same way, many mitigations strategies are available in the literature but only probiotics have been discussed with a few examples. The total weight in the abstract has been put on the probiotics but the information on the subject has been presented very succintly. The title does not represent the study parameters taken for consideration. In addition, the authors have also discussed heat shock protien and thyroid hormones. I do not know why these specific parameters have been chosen. In addition, why intestinal and immune response have been preferred in the title but the other parameters have been ignored. Furthermore, no tabulated data has been given for intestinal ultrastrucutres. I think the authors have tried to go shortcut to publish some review of literature from a thesis where only limited data was available. Moreover, the information provided on each topic is very few and selected. Intrestingly, few information have been presented on mitigation strategy through probiotics, however, exaggerated statements have been given in the conclusion section about it. Proper management has been written several times and much emphasis has been given, however, nothing has been given about proper management. Therefore, in my opnion, this review is very weak in the basic tenants of an ideal review.
Author Response
Revision Notes
This revision notes are disclosed to address comments/reviews on " animals-1870307", entitled: Influence of heat stress on poultry growth performance, intestinal inflammation, and immune function and potential mitigation by probiotics
Introduction:
The authors would like to express their gratitude towards the MDPI Animals for permitting us to revise our manuscript, thereby improving the quality of the revised manuscript before the high standard of this publication channel.
The authors attempted as best as possible to strike a balance conciseness and results reporting. The aim is to create an informative paper for the readers that clearly articulate the key contributions. The remarkable improvements of the revised manuscript are:
Major and comprehensive clarifications have been made based on the comments from the Reviewers, as well as they have been implemented in the corresponding section of the revised manuscript.
The modified/revised parts of the manuscript are
Reviewer 1 comments (Highlighted with blue text)
Comments and Suggestions for Authors:
In the current study, the authors have rviewed "Influence of heat stress on poultry growth performance, intestinal inflammation, and immune function and potential mitigation by probiotics". Already there are a lot of review on the subject. The current study does not add anything to the existing knowledge of heat stress on poultry. Many important rerferences have been ignored. For me it is not understandable why a few parameters have been targeted. In the same way, many mitigations strategies are available in the literature but only probiotics have been discussed with a few examples. The total weight in the abstract has been put on the probiotics but the information on the subject has been presented very succinctly. The title does not represent the study parameters taken for consideration. In addition, the authors have also discussed heat shock protien and thyroid hormones. I do not know why these specific parameters have been chosen. In addition, why intestinal and immune response have been preferred in the title but the other parameters have been ignored. Furthermore, no tabulated data has been given for intestinal ultrastrucutres. I think the authors have tried to go shortcut to publish some review of literature from a thesis where only limited data was available. Moreover, the information provided on each topic is very few and selected. Intrestingly, few information have been presented on mitigation strategy through probiotics, however, exaggerated statements have been given in the conclusion section about it. Proper management has been written several times and much emphasis has been given, however, nothing has been given about proper management. Therefore, in my opnion, this review is very weak in the basic tenants of an ideal review.
Answer: We would like to thank the reviewer for the careful and thorough reading of this manuscript and for the thoughtful comments and constructive suggestions, which help to improve the quality of this manuscript. We have thoroughly checked and improved the technicality of the manuscripts and incorporated the possible suggested comments by the reviewers. Hope now it will be according to the standard of this journal. Our response follows.
Worldwide, the effect of climatic variations has become a great challenge in poultry production. Heat stress is amongst the most significant stressors influencing poultry productivity in hot climate regions, causing substantial economic losses in the poultry industry. These economic losses are speculated to increase in the coming years with the global temperature rise. Moreover, modern poultry strains are more susceptible to high ambient temperature. Heat stress has a waste range of negative effects on the poultry industry; however, our main concern was the physiological response, growth performance, and laying performance, which appeared in the form of reducing feed consumption, body weight gain, egg production, feed efficiency, meat quality, egg quality, and immune response, therefore these parameters were considered in our current review. About the brief description of each and every parameter, it will make this review article lengthy and will be beyond the scope of this journal. However, in our upcoming review articles, we will briefly investigate the adverse impacts of thermal stress on a few basic parameters separately and will also elucidate the molecular mechanisms underlying thermal conditioning and its effects on the acquisition of tolerance to acute heat stress in later life as well as the probiotics mitigating effects. To date, several mitigation procedures have been used to ameliorate the negative impacts of increased temperature; of these, dietary manipulation, which gains a great concern in different regions around the world. These nutritional manipulations mainly included feed additives (natural antioxidants, minerals, probiotics, etc.), feed restriction, feed form, drinking cold water, and others. However, in the large-scale poultry industry, only a few of these strategies are commonly used. In the current review article, we deliberate the practical applications of useful nutritional manipulation such as probiotics which are the cheap and safest source to be used as a feed additive to mitigate the heat load in poultry. The documented information will be useful to poultry producers to improve the general health status and productivity of heat-stressed birds via enhancing stress tolerance, oxidative status, and immune response, and thereby provide valuable recommendations to minimize production losses due to heat stress in particular under the growing global warming crisis.

Reviewer 2 Report
The aim of the manuscript animals-1870307 entitled ‘Influence of heat stress on poultry growth performance, intestinal inflammation, and immune function and potential mitigation by probiotics’’ is relevant with the scope of the journal.
The study is well conceived and written. This review and provides useful information for the poultry farming, focusing on the impact of heat stress on poultry health and growth performance as well as the application of probiotics as a promising approach to alleviate heat stress in poultry.
Please let me congratulate you on the quality of your journal and thank you for giving me the opportunity to contribute as a reviewer.
Comments to the authors:
· L86: poultry welfare
· L88: poultry welfare
· L90: poultry performance
· L91-92: poultry welfare
§ L124-125: rephrase the sentence
§ L128: commercial poultry production
§ L170: 96 laying hens (60 weeks of age)
§ L470: reform the table 3, using less words e.g.
Subtitle of the table: Biological performance under heat stress conditions
Probiotic Bacillus subtilis PB6
broiler chicken
-Increased the population of Bifidobacterium and Lactobacilli in the gut 3×107 CFU/kg
-Increased villus height, crypt depth, and promoted surface area for absorption in duodenum and ileum
Probiotics (Bacillus licheniformis)
- Restored the impaired villus-crypt structure and maintained normal surface area in the small intestine (laying hens)
- Avoided the impairment and regulated the stability of the epithelial intestine in (layer chickens).
Probiotic mixture (B. licheniformis, B. subtilis, L. Plantarum)
broilers chickens
- Reversed the reduced villus height, crypt depth, and surface area structure
- Increased beneficial bacterial population Lactobacillus and Bifidobacterium and decreased injurious bacterial population Coliforms in small intestine
Prebiotic mixture (L. Pentosus ITA23 and L. Acidophilus ITA44)
broilers chickens
- Increased the population of beneficial bacteria Lactobacillus, Bifidobacterium, Enterococcus
- Decreased the population of E. coli respectively in the gut
§ Add information in the part of ‘’Discussion’’ about the economic impact of probiotics use in poultry production for the control of heat stress
Author Response
Revision Notes
This revision notes are disclosed to address comments/reviews on " animals-1870307", entitled: Influence of heat stress on poultry growth performance, intestinal inflammation, and immune function and potential mitigation by probiotics
Introduction:
The authors would like to express their gratitude to the MDPI Animals for permitting us to revise our manuscript, thereby improving the quality of the revised manuscript before the high standard of this publication channel.
The authors attempted as best as possible to strike balance between conciseness and results reporting. The aim is to create an informative paper for the readers that clearly articulate the key components. The remarkable improvements of the revised manuscript are:
Major and comprehensive clarifications have been made based on the comments from the Reviewers, as well as they have been implemented in the corresponding section of the revised manuscript.
The modified/revised parts of the manuscript are
Reviewer 2 Comments (Highlighted with green text)
The aim of the manuscript animals-1870307 entitled ‘Influence of heat stress on poultry growth performance, intestinal inflammation, and immune function and potential mitigation by probiotics’’ is relevant to the scope of the journal.
The study is well conceived and written. This review and provides useful information for poultry farming, focusing on the impact of heat stress on poultry health and growth performance as well as the application of probiotics as a promising approach to alleviate heat stress in poultry.
Please let me congratulate you on the quality of your journal and thank you for giving me the opportunity to contribute as a reviewer.
Answer: Thank you for your very careful review of our paper, and for the comments and suggestions that ensued.
Comments to the authors:
- L86: poultry welfare
Answer: The authors would like to thank the reviewer for the careful and thorough reading of this manuscript and for the thoughtful comments and constructive suggestions, which help to improve the quality of this manuscript. We have thoroughly checked and improved the technicality of the manuscripts and incorporated the possible suggested comments by Reviewer 2. According to the Reviewers suggestion following changes have been made in the revised version of the manuscript.
Avian welfare has been replaced with poultry welfare in L86 and L88 and chicken performance has been replaced with poultry performance in L90 and avian well-being has been replaced with poultry welfare in L91-92.
- L124-125: rephrase the sentence
Answer: The lines have been rephrased as follows in the revised version. In addition, poultry under heat stress conditions has reduced feed intake, water consumption, and body moments, and consequently become depressed, dull, and lethargic [29].
- L128: commercial poultry production
Answer: Commercial chicken production has been replaced with commercial poultry production in L128
- L170: 96 laying hens (60 weeks of age)
Answer: ninety-six 60-week old laying hens have been replaced with 96 laying hens (60 weeks of age) in L170
- L470: reform the table 2, using less words e.g. Subtitle of the table: Biological performance under heat stress conditions
Answer: Biological performance has been replaced with Biological performance under heat stress conditions in L470 in revised table 2.
Probiotic Bacillus subtilis PB6 broiler chicken
Answer: Probiotic Bacillus subtilis PB6, (3x107 CFU/kg) has been replaced by Probiotic Bacillus subtilis PB6 broiler chicken in the revised table 2. Increased the population of Bifidobacterium and Lactobacilli in the gut of broiler chickens under heat stress environment. Increased villus height, crypt depth, and promoted surface area for absorption in duodenum and ileum of broiler chicken under heat stress conditions has been replaced by increased the population of Bifidobacterium and Lactobacilli in the gut 3×107 CFU/kg. Increased villus height, crypt depth, and promoted surface area for absorption in duodenum and ileum in the revised table 2. Restored the impaired villus-crypt structure and maintained normal surface area in the small intestine of laying hen reared under heat stress conditions. Avoided the impairment and regulated the stability of the epithelial intestine in layer chickens subjected to heat stress conditions. The extreme rise in number and computation of mast cells endure and secures mucin-secreting cells, hence, the intraluminal defensive barrier mechanism becomes more developed have been replaced by Restored the impaired villus-crypt structure and maintained normal surface area in the small intestine (laying hens). Avoided the impairment and regulated the stability of the epithelial intestine in (layer chickens) in the revised table 2.
Probiotic mixture (B. licheniformis, B. subtilis, L. Plantarum) has been replaced by Probiotic mixture (B. licheniformis, B. subtilis, L. Plantarum) broilers chickens in the revised table 2.
Reversed the reduced villus height, crypt depth, and surface area structure. Increased beneficial bacterial population Lactobacillus and Bifidobacterium and decreased injurious bacterial population Coliforms in small intestine of broiler chickens raised under heat stress conditions have been replaced by Reversed the reduced villus height, crypt depth, and surface area structure.
Increased beneficial bacterial population Lactobacillus and Bifidobacterium and decreased injurious bacterial population Coliforms in small intestine in the revised table 2.
Prebiotic mixture (L. Pentosus ITA23 and L. Acidophilus ITA44) has been replaced by Prebiotic mixture (L. Pentosus ITA23 and L. Acidophilus ITA44) broilers chickens in the revised table 2.
Increased the population of beneficial bacteria Lactobacillus, Bifidobacterium, Enterococcus, and decreased the population of E. coli respectively in the gut of broilers chickens due to heat stress conditions has been replaced by Increased the population of beneficial bacteria Lactobacillus, Bifidobacterium, Enterococcus Decreased the population of E. coli respectively in the gut in the revised table 2.
- Add information in the part of ‘’Discussion’’ about the economic impact of probiotics use in poultry production for the control of heat stress
Answer: Cost about application of probiotics in poultry in response to heat stress. Nutritional approaches can balance environmental distress during persistent heat stress. Feed additives have been supplemented to diets as part of a nutritional approach to mitigate the negative effects of heat stress in poultry [17]. Probiotics increase the nutritive value of feed by converting less degradable compounds into more digestible forms. As a result, the gut microbial population shifts, the composition of the gut microbiota changes, and the immune system is modulated. Feed digestion is improved by producing hydrolytic enzymes (phytases, lipases, amylases, or proteases) that enhance nutrient intake or by stimulating the host to generate more digestive enzymes [146]. Probiotics may enhance the production of vitamins, exopolysaccharides, and antioxidants, thereby boosting the nutritional value of feed. Some probiotics may also regulate cholesterol metabolism [147]. The digestion and absorption of nutrients take place in the proventriculus and small intestines. Ileum and colon boost up the fermentation of undigested proteins and carbohydrates. This is a critical step in digestion because undigested nutrients are a risk factor for dysbiosis [148].

Reviewer 3 Report
Dear Authors,
Thank you for submitting this interesting article investigating heat stress in chickens and the remedial effects of probiotics. I found your work to be engaging and in a world where climate change is a reality, there is real value to this work.
At current however, there seem to be some large revisions required in the manuscript to ensure the work is scientifically robust. I have attached the PDF version of the manuscript with specific comments. Additionally, please consider the following points:
1. There are several generalisations made in the work regarding how bird bodies are impacted by heat stress. Please be specific about which species is being discussed. This is particularly important as a wide array of species are under discussion (e.g. chickens and turkeys which are Galliformes, and ducks which are Anseriformes) so there is potential that different taxa may respond differently. Please also be clear as to what is being discussed in this review - which species are covered?
2. Include scientific names when discussing species for the first time in text. This is essential to prevent confusion regarding species.
3. Please cite every point that is not common knowledge. Without this, it is difficult to see where information originated.
4. Climate change. The climate change point is key for your manuscript, and gives it real currency. Please provide some more details on how the climate is predicted to change and the real world impact this will play on your species. This is covered in minimal detail at current.
5. Future applications. What are the best options for those involved in farming of poultry? Please draw together a section consisting of future study areas and implications to minimise heat stress effects.

Author Response
Revision Notes
This revision notes are disclosed to address comments/reviews on " animals-1870307", entitled: Influence of heat stress on poultry growth performance, intestinal inflammation, and immune function and potential mitigation by probiotics
Introduction:
The authors would like to express their gratitude to the MDPI Animals for permitting us to revise our manuscript, thereby improving the quality of the revised manuscript before the high standard of this publication channel.
The authors attempted as best as possible to strike balance between conciseness and results reporting. The aim is to create an informative paper for the readers that clearly articulate the key components. The remarkable improvements of the revised manuscript are:
Major and comprehensive clarifications have been made based on the comments from the Reviewers, as well as they have been implemented in the corresponding section of the revised manuscript.
The modified/revised parts of the manuscript are
Reviewer 3 Comments (Highlighted with purple text)
Comment 1: Thank you for submitting this interesting article investigating heat stress in chickens and the remedial effects of probiotics. I found your work to be engaging and, in a world, where climate change is a reality, there is real value to this work.
Answer: Thank you for your very careful review of our paper, and for the comments and suggestions that ensued.
At current, however, there seem to be some large revisions required in the manuscript to ensure the work is scientifically robust. I have attached the PDF version of the manuscript with specific comments. Additionally, please consider the following points:
- There are several generalizations made in the work regarding how bird bodies are impacted by heat stress. Please be specific about which species is being discussed. This is particularly important as a wide array of species are under discussion (e.g. chickens and turkeys which are Galliformes, and ducks which are Anseriformes) so there is potential that different taxa may respond differently. Please also be clear as to what is being discussed in this review - which species are covered?
Answer: The authors would like to thank the reviewer for the careful and thorough reading of this manuscript and for the thoughtful comments and constructive suggestions, which help to improve the quality of this manuscript. We have thoroughly checked and improved the technicality of the manuscripts and incorporated the possible suggested comments by Reviewer 3. According to the Reviewers suggestion following changes have been made in the revised version of the manuscript. Heat stress is considered a critical obstacle coping with poultry farming in hot climate areas, triggering major economic losses in the poultry industry. Heat stress starts once the ambient temperature rises above the comfort zone (16°C -25°C) for poultry species and (36°C-41°C) for ducks. Along with the excessive rise in the ambient temperature generally and the average rise in the global temperature specifically, relative humidity also contributes to the development of heat stress. In this article, several generalizations have been made in the work regarding how bird bodies are impacted by heat stress. Above the comfort zone, poultry, duck and turkey share a common mechanism of heat dissipation from the body through panting. In this review article, ducks, turkeys, and quails are covered generally and poultry specifically.
Akbarian, A.; Michiels, J.; Degroote, J.; Majdeddin, M.; Golian, A.; De Smet, S. Association between heat stress and oxidative stress in poultry; mitochondrial dysfunction and dietary interventions with phytochemicals. J. Anim. Sci. Biotechnol. 2016, 7 (1), 1-14.
Further detail about poultry, duck, turkeys, and quail have been covered in Table 1 regarding heat stress effects along with temperature conditions, outcomes and references.
- Include scientific names when discussing species for the first time in text. This is essential to prevent confusion regarding species.
Answer: Thanks for the valuable suggestion. This will make our review article more valuable and scientific. The scientific name has been included in the revised manuscript.
- Please cite every point that is not common knowledge. Without this, it is difficult to see where the information originated.
Answer: Thanks for the valuable suggestion. The necessary mention citation has been updated.
- Climate change. The climate change point is key for your manuscript and gives it real currency. Please provide some more details on how the climate is predicted to change and the real-world impact this will play on your species. This is covered in minimal detail at current.
Answer: The Intergovernmental Panel on Climate Change reported a 1.53 °C increase in average environmental temperature between 2006-2015 compared to 1850-1900. It has nearly doubled in comparison to the average global temperatures since the pre-industrial period. This steady increase was anticipated to result in novel, hot climates, particularly in tropical regions [1]. Climate change has an adverse impact on food security because of its detrimental consequences on agricultural crops and livestock [2]. Furthermore, the temperature is regarded as the most important environmental determinant among all bioclimatic variables known to cause stress. Nowadays, heat stress is one of the major climatic issues affecting the production, reproduction, and growth performance of various livestock species and poultry [3]. Poultry under high ambient temperatures, develop physiological, behavioural, and immunological responses which directly or indirectly adversely affect their performance [4].
- Future applications. What are the best options for those involved in farming of poultry? Please draw together a section consisting of future study areas and implications to minimize heat stress effects.
Answer: However, further research is needed to investigate the molecular changes induced by probiotics as well as the interactions between probiotics, pathogens, and epithelial cells. This will necessarily involve metagenomic, proteomic, and metabolomic exploration. Elaborating these unknowns will give researchers a stronger insight into the role probiotics play in improving poultry health and growth.
- Simple summary is quite brief. Please expand on the key findings from the review. Poultry industry sustains severe economic loss under heat stress conditions. Heat stress adversely affects the productivity, physiological status, and immunity of birds. To date, several mitigation measures have been adopted to minimize the negative effects of heat stress in poultry. Nutritional strategies have been explored as a promising approach to mitigate heat stress-associated deleterious impacts. Of these, probiotic feeding is the main nutritional strategy, and this approach warrants further investigation to improve thermotolerance in poultry.
- Some of the key words are already included in the title. Remove any key words that are in the title and use new terms to increase paper discoverability
Answer: Thermoregulation; reproduction; strategies; body temperature; poultry.
- include scientific name on first mention
Answer: Gallus gallus domesticus.
- What are these losses from? Heat stress?
Answer: These losses are from heat stress. As heat stress have detrimental affects on the welfare of poultry, as a result, poultry are unable to maintain normal survival life.
- Could you provide some more information on this as it is key to your arguments. How much is the temperature expect to increase by year X? Is there any estimation of impact on poultry?
Answer: The Intergovernmental Panel on Climate Change reported a 1.53 °C increase in average environmental temperature between 2006-2015 compared to 1850-1900. It has nearly doubled in comparison to the average global temperatures since the pre-industrial period. This steady increase was anticipated to result in novel, hot climates, particularly in tropical regions [1]. Climate change has an adverse impact on food security because of its detrimental consequences on agricultural crops and livestock [2]. Furthermore, the temperature is regarded as the most important environmental determinant among all bioclimatic variables known to cause stress. Nowadays, heat stress is one of the major climatic issues affecting the production, reproduction, and growth performance of various livestock species and poultry [3]. Poultry under high ambient temperatures, develop physiological, behavioural, and immunological responses which directly or indirectly adversely affect their performance [4].
References
- IPCC. Climate Change and Land: an IPCC special report on climate change, desertification, land degradation, sustainable land management, food security, and greenhouse gas fluxes in terrestrial ecosystems. 2019.
- Shukla, P. R.; Skeg, J.; Buendia, E. C.; Masson-Delmotte, V.; Pörtner, H.-O.; Roberts, D.; Zhai, P.; Slade, R.; Connors, S.; Van Diemen, S. Climate Change and Land: an IPCC special report on climate change, desertification, land degradation, sustainable land management, food security, and greenhouse gas fluxes in terrestrial ecosystems. 2019.
- Alagawany, M.; Farag, M.; Abd El-Hack, M.; Patra, A. Heat stress: effects on productive and reproductive performance of quail. Worlds Poult Sci. J. 2017, 73 (4), 747-756.
- Nawab, A.; Ibtisham, F.; Li, G.; Kieser, B.; Wu, J.; Liu, W.; Zhao, Y.; Nawab, Y.; Li, K.; Xiao, M. Heat stress in poultry production: Mitigation strategies to overcome the future challenges facing the global poultry industry. J. Therm. Biol. 2018, 78, 131-139.
- Not biologically accurate - chickens can release heat even if they don't have sweat glands.
Answer: Chicken can also release heat through panting, but excessive panting leads to respiratory alkalosis and deficiency of calcium in the body, consequently lameness develops.
- This needs to be cited.
Answer: Kadykalo, S.; Roberts, T.; Thompson, M.; Wilson, J.; Lang, M.; Espeisse, O. The value of anticoccidials for sustainable global poultry production. Int. J. Antimicrob. Agents. 2018, 51 (3), 304-310.
- and well-being?
Answer: corrected.
- citation is missing
Answer: Zhang, Y.; Proenca, R.; Maffei, M.; Barone, M.; Leopold, L.; Friedman, J. M. Positional cloning of the mouse obese gene and its human homologue. Nature 19.94, 372 (6505), 425-432.
- citation missing
Answer: Zhang, Y.; Proenca, R.; Maffei, M.; Barone, M.; Leopold, L.; Friedman, J. M. Positional cloning of the mouse obese gene and its human homologue. Nature 1994, 372 (6505), 425-432.
- Include scientific name on first mention
Answer: Mus musculus.
- check sentence structure here
Answer: Corrected in the revised manuscript.
- Include scientific names on first mention
Answer: quails (Coturnix coturnix), hens Gallus gallus domesticus, and ducks Anatidae.
- Include scientific names on first mention
Answer: Coturnix japonica.
- As measured by?
Answer: Corrected in the revised manuscript.
- cite here
Answer: Türk, G.; Çeribaşı, A. O.; ÅžimÅŸek, Ü. G.; Çeribaşı, S.; Güvenç, M.; Kaya, Åž. Ö.; Çiftçi, M.; Sönmez, M.; Yüce, A.; Bayrakdar, A. Dietary rosemary oil alleviates heat stress-induced structural and functional damage through lipid peroxidation in the testes of growing Japanese quail. Anim. Reprod. Sci. 2016, 164, 133-143.
- Out of how many?
Answer: Five Japanese male quails (Coturnix japonica) out of 45 Japanese male quails have been updated in the revised manuscript.
- shelduck? Please also include scientific name here
Answer: Tadorna ferrugineasss.
- How much less? What was the effect size?
Answer: Decrease T3 (37.5%) Increase T4 (30%).
- Cite here
Answer: Galarza-Seeber, R.; Latorre, J. D.; Bielke, L. R.; Kuttappan, V. A.; Wolfenden, A. D.; Hernandez-Velasco, X.; Merino-Guzman, R.; Vicente, J. L.; Donoghue, A.; Cross, D. Leaky gut and mycotoxins: Aflatoxin B1 does not increase gut permeability in broiler chickens. Front. Vet. Sci. 2016, 3, 10.
- is this section related specifically to chickens, then?
Answer: There is overall description about the immune system of duck, quail, turkey, and poultry. However, poultry is the main species in the overall review article.
- cite here
Answer: Goel, A. Heat stress management in poultry. J Animal Physiol Anim Nutr 2021, 105 (6), 1136-1145.
- cite here
Answer: Infante, M.; Armani, A.; Mammi, C.; Fabbri, A.; Caprio, M. Impact of adrenal steroids on regulation of adipose tissue. Compr. Physiol. 2011, 7 (4), 1425-1447.
- of all species? cite source.
Answer: animal has been replaced by poultry in the revised manuscript.
- Scientific names
Answer: Gallus gallus domesticus.
- if there are numerous researchers, please cite them.
Answer: Al-Fataftah, A.-R.; Abdelqader, A. Effects of dietary Bacillus subtilis on heat-stressed broilers performance, intestinal morphology and microflora composition. Animal feed science and technology 2014, 198, 279-285.
- with which probiotics?
Answer: (mannan-oligosaccharide and Lactobacillus)
Reference. Sohail, M. U.; Ijaz, A.; Yousaf, M.; Ashraf, K.; Zaneb, H.; Aleem, M.; Rehman, H. Alleviation of cyclic heat stress in broilers by dietary supplementation of mannan-oligosaccharide and Lactobacillus-based probiotic: Dynamics of cortisol, thyroid hormones, chole[5]sterol, C-reactive protein, and humoral immunity. Poult. Sci. 2010, 89 (9), 1934-1938.
- Could you build on this point as a future applications / directions subheading.
Answer: However, further research is needed to investigate the molecular changes induced by probiotics as well as the interactions between probiotics, pathogens, and epithelial cells. This will necessarily involve metagenomic, proteomic, and metabolomic exploration. Elaborating these unknowns will give researchers a stronger insight into the role probiotics play in improving poultry health and growth.
- Is this just one page? Should the volume be in italics for MDPI?
Answer: All the references has been arranged and written according to MDPI Animal template style.
- Check page numbers in journal references for consistency
Answer: Zaheer, K. An updated review on chicken eggs: production, consumption, management aspects and nutritional benefits to human health. Food Sci. Nutr. 2015, 6 (13), 1208-1220.
- Volume, pages?
Answer: c.
- Does the author really have six names?
Answer: Yes, the author have six names.
- Should be capitalised
Answer: Corrected.
- Italicise the scientific name
Answer: Corrected.
- Italicise the scientific name
Answer: Corrected.
- Italicise the scientific name
Answer: Corrected.
- Italicise the scientific name
Answer: Corrected.
- Italicise the scientific name
Answer: Corrected.
- Italicise the scientific name
Answer: Corrected.

Round 2
Reviewer 1 Report
In my first review report, I had rejected this review for not being able to compete with other available reviews already present in the literature. I am still stick to my view points and just cosmetic changes from the authors does not qualify it to be accepted in the present form.
Author Response
Comments and Suggestions for Authors:
In my first review report, I had rejected this review for not being able to compete with other available reviews already present in the literature. I am still stick to my view points and just cosmetic changes from the authors does not qualify it to be accepted in the present form.
Answer: Thank you for your very careful review of our paper, and for the comments and suggestions that ensued.

Reviewer 3 Report
Dear Authors,
Many thanks for submitting this revised version of the manuscript for review. You have taken into account the feedback provided on the initial review of the paper. You have also shown clearly where changes have been made to the work. The developments to the manuscript have resulted in a more robust paper overall. In light of the revisions, the paper is now in a much better position for consideration.
Author Response
Comments and Suggestions for Authors:
Many thanks for submitting this revised version of the manuscript for review. You have taken into account the feedback provided on the initial review of the paper. You have also shown clearly where changes have been made to the work. The developments to the manuscript have resulted in a more robust paper overall. In light of the revisions, the paper is now in a much better position for consideration.
Answer: Thank you for your very careful review of our paper, and for the comments and suggestions that ensued.
